Heterogeneous text graph for comprehensive multilingual sentiment analysis: capturing short- and long-distance semantics

Mercha El Mahdi 1 2 elmahdi.mercha@um5s.net.ma
Benbrahim Houda 1
Erradi Mohammed 1
1 ENSIAS, Mohammed V University in Rabat , Rabat , Morocco
2 HENCEFORTH , Rabat , Morocco
Alatas Bilal
Electronic publication date: 2024 Feb 23
Publication date: 2024
Volume: 10
Electronic Location ID: e1876
Received 2023 Oct 11; Accepted 2024 Jan 22
Copyright: © 2024 Mercha et al.
Copyright year: 2024
Copyright holder: Mercha et al.
License: This is an open access article distributed under the terms of the Creative Commons Attribution License, which permits unrestricted use, distribution, reproduction and adaptation in any medium and for any purpose provided that it is properly attributed. For attribution, the original author(s), title, publication source (PeerJ Computer Science) and either DOI or URL of the article must be cited.
License URL: https://creativecommons.org/licenses/by/4.0/

Keywords: Multilingual sentiment analysis, Graph convolutional network, Natural language processing, Information retrieval

Funding: The authors received no funding for this work.

==============================
Multilingual sentiment analysis (MSA) involves the task of comprehending people’s opinions, sentiments, and emotions in multilingual written texts. This task has garnered considerable attention due to its importance in extracting insights for decision-making across diverse fields such as marketing, finance, and politics. Several studies have explored MSA using deep learning methods. Nonetheless, a majority of these studies depend on sequential-based approaches, which focus on capturing short-distance semantics within adjacent word sequences, but they overlook long-distance semantics, which can provide more profound insights for analysis. In this work, we propose an approach for multilingual sentiment analysis, namely MSA-GCN, leveraging a graph convolutional network to effectively capture both short- and long-distance semantics. MSA-GCN involves the comprehensive modeling of the multilingual sentiment analysis corpus through a unified heterogeneous text graph. Subsequently, a slightly deep graph convolutional network is employed to acquire predictive representations for all nodes by encouraging the transfer learning across languages. Extensive experiments are carried out on various language combinations using different benchmark datasets to assess the efficiency of the proposed approach. These datasets include Multilingual Amazon Reviews Corpus (MARC), Internet Movie Database (IMDB), Allociné, and Muchocine. The achieved results reveal that MSA-GCN significantly outperformed all baseline models in almost all datasets with a p-value < 0.05 based on student t-test. In addition, such approach shows prominent results in a variety of language combinations, revealing the robustness of the approach against language variation.

Introduction

Sentiment analysis plays a crucial role across various sectors that significantly impact our lives, including finance, marketing, politics, and security. This importance can be attributed to its capacity in retrieving valuable insights for informed decision-making. Sentiment analysis is viewed as a computational study of sentiments, emotions, opinions, and appraisals to better understand a person’s reactions and attitudes, towards several entities, as elaborated in the previous works (Liu, 2020; Mercha & Benbrahim, 2023). Although sentiment analysis has received extensive attention and seen the development of several approaches, the majority of these approaches concentrate on constructing monolingual systems. However, people worldwide tend to express themselves in both their native languages and foreign languages, leading to multilingual content. Therefore, analyzing sentiments in a single language increases the risk of overlooking important information contained within texts written in other languages.

Since traditional monolingual sentiment analysis systems are ineffective to handle the newly generated multilingual content, multilingual sentiment analysis (MSA) is becoming an increasingly important topic of study. Several studies have been proposed in the literature for MSA, which aim to develop language-independent and translation-free systems (Agüero-Torales, Salas & López-Herrera, 2021). MSA research ranges from establishing the best features to selecting the most adequate machine learning and deep learning classifiers. Most MSA approaches rely on deep learning architectures, particularly convolutional neural network (CNN) (Kim, 2014) and long short-term memory (LSTM) (Wang et al., 2015), to learn insightful information from word or character features (Mercha & Benbrahim, 2023). While these deep learning architectures are generally adept at capturing short-distance semantics within local consecutive word sequences, they miss global word co-occurrence across the entire corpus, which is essential for conveying long-distance semantics (Peng et al., 2018). However, considering long-distance semantics in sentiment analysis is very crucial for achieving a more nuanced and contextually accurate understanding of the sentiments.

In recent years, graph neural networks (GNNs) have gained considerable attention, as they can learn efficient representations over dependencies between entities in a graph. GNNs are widely considered as effective and flexible techniques for graph representations learning, with successful applications in a wide range of tasks, including text classification (Zhang et al., 2020), social network analysis (Zhang et al., 2022), and drug discovery (Gaudelet et al., 2021).

In this work, we propose a multilingual sentiment analysis approach based on graph convolutional network, which captures both short- and long-distance semantics by considering the global word co-occurrence across the entire corpus. We establish a unified heterogeneous text graph to represent the entire data of a multilingual corpus. Nodes in the graph correspond to words and documents, and edges are formed by incorporating sequential, semantic, and statistical information. Employing a slightly deep graph convolutional network, we model the entire graph, leveraging heterogeneous information to learn representations for both words and documents. The acquired document representations are then utilized for sentiment prediction. The proposed approach is capable of capturing both short- and long-distance semantics that are extremely important for the analysis. Moreover, the experiments reveal the prominent results of the MSA-GCN approach across various language combinations, highlighting its resilience to language variations.

In this article, we extend our previous work (Mercha, Benbrahim & Erradi, 2023) with additional experimentation to assess the efficiency of the proposed approach. To the best of our knowledge, this work is the first comprehensive study which explore graph neural network for multilingual sentiment analysis. The main contributions are summarized as follows: We represent the whole multilingual corpus with a single heterogeneous text graph, based on semantic, sequential and statistical information to consider both short- and long-distance semantics.

We propose a slightly deep graph convolutional network to learn predictive representations.

We carried out a wide range of experiments using different language combinations to assess the effectiveness and resilience of the general approach when confronted with language variations.

The remainder of this article is structured as follows: “Related Works” presents the major approaches developed for multilingual sentiment analysis and graph neural networks. “The Proposed Approach” clearly describes the proposed approach starting from the graph construction to the learning of predictive nodes representations. “Experiments” describes the experiment’s settings and discusses the findings of the proposed approach. “Conclusion and Future Works” provides the conclusion and future works.

Related works

In this section, we provide an overview about the developments achieved in conducting multilingual sentiment analysis based on machine learning and deep learning approaches. These approaches are categorized according to the methodology adopted for the representation learning. Additionally, we outline advancements made in graph neural networks. Subsequently, we explore the constraints of current approaches and propose a novel approach that leverages the capabilities of graph neural networks.

Multilingual sentiment analysis

Given the great importance of multilingual sentiment analysis in today’s interconnected world, several studies have been directed towards advancing methodologies, and enhancing accuracy to better perform sentiment analysis across languages, as previously introduced in Mercha & Benbrahim (2023). These studies aim to build language-agnostic systems that can learn predictive representations directly from multilingual content to predict the sentiments. The approaches outlined in the literature can be categorized into two approaches, depending on the methodologies used for text representation: word-based and character-based.

Previous research studies on word-based approaches investigated several classical machine learning classifiers with various handcrafted features, such as n-grams and bag-of-words (Narr, Hulfenhaus & Albayrak, 2012; Kincl et al., 2015; Abudawood, Alraqibah & Alsanie, 2018). In the study (Narr, Hulfenhaus & Albayrak, 2012), a suggestion was made to train naïve Bayes classifier on the generated multilingual data by a semi-supervised heuristic labeling method. The trained classifier was then tested on human annotated tweets using different n-gram features. The outcomes obtained from these experiments indicated that training a classifier using a mixture of multiple languages remained feasible, despite a slight decrease in performance. Kincl et al. (2015) introduced an approach that enhances the predictive capabilities of the support vector machine classifier by integrating the sentiment from local and chronological surrounding context. The achieved results across diverse datasets covering various languages and domains demonstrated the effectiveness of the proposed approach. Abudawood, Alraqibah & Alsanie (2018) suggested language-independent sentiment analysis approach which relies on word statistics and structural co-occurrence. The achieved results suggest that the proposed approach is better suited for online sentiment. Pribán & Balahur (2019) introduced and assessed three sentiment analysis systems designed to address three distinct sentiment analysis tasks within a multilingual context. The results of the experiments proved the proficiency of these systems within the context of multilingual capabilities.

Recently, the development of deep learning techniques has led to the widespread use of CNN and LSTM in MSA. Several studies, such as Medrouk & Pappa (2017), Deriu et al. (2017) and Attia et al. (2018), designed architectures based on CNN and randomly initialized word embeddings which learn directly from multilingual data. Medrouk & Pappa (2018) investigated CNN and LSTM in the multilingual context without any special feature considerations. The experimental results proved the effectiveness of the proposed architectures in both monolingual and multilingual settings, all without adopting any linguistic resources. Shakeel et al. (2020) proposed a hybrid architecture combining CNN and LSTM models to capture both n-gram features and long-term dependencies. The results indicated the model’s resilience against class label distribution skewness and its ability to maintain robust performance across various mixed languages. Manias et al. (2023) investigated four multilingual Transformer-based classifiers and zero-shot classification approach in MSA task. The experimental results revealed that Multilingual BERT-based classifiers achieved high performance and transfer inference when trained on multilingual data.

While word-based approaches are widely employed for handling MSA, numerous studies have introduced character-based approaches as an alternative. Most character-based methods leverage convolutional neural networks to construct efficient architectures that can extract valuable insights from character features (Becker et al., 2017; Zhang, Zhang & Chan, 2017a; Wehrmann et al., 2017; Wehrmann, Becker & Barros, 2018). In Becker et al. (2017), a translation-free deep neural approach for MSA was introduced, where three different neural architectures were compared using tweets from four distinct languages. The obtained results demonstrated that the suggested architecture was competitive with the baseline models, offering a notable improvement by decreasing the number of parameters and memory usage. Similarly, Zhang, Zhang & Chan (2017a) proposed a language-agnostic model, namely unicode character convolutional neural network (UniCNN), designed for sentiment classification. UniCNN was assessed on six Twitter datasets from distinct languages, and the outcomes demonstrated its superior performance over state-of-the-art models. In the study conducted in Wehrmann et al. (2017), a series of experiments were carried out to investigate the relationship between the performance of their proposed character-based CNN model and various architectural parameters. The findings underscored the efficiency of these models in multilingual contexts. Wehrmann, Becker & Barros (2018) described a multilingual character-based approach that simultaneously addresses language detection and sentiment analysis. Extensive experiments were conducted on tweets from various languages, and the results highlighted the superiority of the proposed approach compared to the baseline models. In a departure from these studies, Zhang, Zhang & Chan (2017b) proposed an approach that combines word and character features. The experimental results demonstrated the effectiveness of fusing deep features from both character and word levels within a CNN architecture.

Graph neural networks

Graph neural networks have gained considerable attention because of its convincing performance. GNNs were widely explored in various areas of natural language processing, including text classification (Yao, Mao & Luo, 2019), sentiment analysis (Li et al., 2022), information extraction (Li et al., 2023), and knowledge graph (Xia et al., 2023).

Earlier studies in the field of graph analysis typically rely on classical machine learning algorithms and hand-crafted features which are time-consuming and performance-limited. Motivated by the success of the methods developed for representation learning (Mikolov et al., 2013a, 2013b), graph representation learning methods have been invented to overcome the limitations of the traditional approaches (Zhang et al., 2018; Cai, Zheng & Chang, 2018). The goal of graph representation learning methods is to develop deep learning architectures that can combine information from the graph topologies and attributes into low-dimensional embeddings (Hamilton, Ying & Leskovec, 2017b). Several graph representation learning algorithms have been developed to explore deep semantic connections in the graph, such as GAT (Veličković et al., 2018), GraphSage (Hamilton, Ying & Leskovec, 2017a), GCN (Kipf & Welling, 2016), and GGNN (Li et al., 2016).

The majority of the suggested approaches in existing literature for multilingual sentiment analysis predominantly stem from either traditional machine learning or deep learning methods. Traditional machine learning approaches depend on statistical representation and employ various feature selection methods, but they suffer from a deficiency in semantic information, posing a significant challenge for sentiment analysis. On the other hand, deep learning-based methods rely on randomly initialized word embeddings or sparse character representations in conjunction with deep learning architectures. These approaches prioritize capturing short-distance semantics within local consecutive word sequences, neglecting the global co-occurrence of words throughout the entire corpus, which is crucial for conveying long-distance semantics.

In contrast to the aforementioned studies, in this work, we propose a novel approach to multilingual sentiment analysis that takes into account both short- and long-distance semantics to enrich the analytical process with deeper insights. This is accomplished by constructing a unified heterogeneous text graph for the entire multilingual corpus, integrating statistical, sequential, and semantic information. Subsequently, a slightly deep graph convolutional network is employed to acquire predictive representations by encouraging the transfer learning across languages.

The proposed approach

In this section, we will go through the proposed approach in detail. First, we show the method of building a single heterogeneous text graph to model the entire multilingual corpus based on sequential, semantic, and statistical information. Then, we describe a slightly deep graph convolutional network to model the graph and to learn efficient words and documents representations. Finally, we introduce the process of evaluating the classification performance of the approach. The overall architecture of the proposed approach is shown in Fig. 1.

Figure 1 The architecture of the proposed MSA-GCN approach.

A slightly deep graph convolutional network learns effective words and documents representations on the constructed heterogeneous text graph. The learned documents representations are then used to predict the sentiment.

Heterogeneous text graph construction

To represent the complete multilingual corpus, we create a unified heterogeneous text graph that incorporates both words and documents as nodes. The graph consists of two types of edges: word-document edges and word-word edges, connecting the respective nodes together. The word-document edges are constructed using statistical information. The weight of an edge between a word node and a document node is calculated using the term frequency-inverse document frequency (tf-idf) measure, which indicates the significance of a word within a document in the corpus. Therefore, the weight of edge connecting a word and a document is determined by the following formula:

(1) tf−idf(w,d,C)=tf(w,d)×idf(w,C)

where tf(w,d) and idf(w,C) denote respectively the relative frequency of word w within document d and the logarithmically scaled inverse fraction of documents of the corpus C containing the word w.

In addition, the word-word edges are established by considering two different aspects of language information: sequential and semantic properties. For capturing sequential relationships, we employ a sliding window approach to gather statistics on word co-occurrence across the entire corpus. The weight between two word nodes is then determined using the measure of positive pointwise mutual information (PPMI) (Church & Hanks, 1990). Mathematically, the weight of a word pair wi, wj is calculated as follows:

(2) PPMI(wi,wj)=max(logp(wi,wj)p(wi)p(wj),0)

(3) p(wi,wj)=#W(wi,wj)#W

(4) p(wi)=#W(wi)#W

where #W(wi,wj) is the number of sliding windows in which both words wi and wj co-occurred, #W(wi) is the number of sliding windows containing the word wi, and #W is the total number of the sliding windows in the whole corpus.

Relying just on sequential information to establish word-word relationships may lead to a disconnected graph structure. To address this, we incorporate semantic information to connect languages and create a unified, interconnected graph. As the proposed MSA-GCN approach is language-agnostic, we adopt the method proposed by Conneau et al. (2017) to align monolingual word embedding spaces in an unsupervised manner. The method utilizes adversarial training to learn a linear mapping W∗ between two languages and it consists of two steps. Initially, a discriminator is trained to differentiate between word embeddings that have been mapped from the two languages, while the mapping is trained to prevent the discriminator from making correct predictions. In the second step, a synthetic dictionary is derived from the resulting shared embedding space, which is then used to refine the mapping through the closed-form Procrustes solution (Schönemann, 1966). For more details, let us assume that we have two languages, L1 and L2, each one is associated with a set of word embedding, X={x1,...,xn} and Y={y1,...,ym} respectively, trained independently on monolingual data. The discriminator is trained to distinguish between elements randomly sampled from W∗X={W∗x1,...,W∗xn} and Y. However, W∗ is trained to make W∗X and Y similar as possible. The discriminator loss LD and the mapping loss LW are defined as follows:

(5) LD(θD|W∗)=−1n∑i=1nlogPθD(L1|W∗xi)−1m∑i=1mlogPθD(L2|yi)

(6) LW∗(W∗|θD)=−1n∑i=1nlogPθD(L2|W∗xi)−1m∑i=1mlogPθD(L1|yi)

where θD is the discriminator parameters and PθD(Li|z) denotes the probability that a vector z is the mapping of word embedding from Li. During the training process, the discriminator and the mapping are learned to respectively minimize LD and LW∗ using the training procedure of deep adversarial networks (Goodfellow et al., 2014). The learned mapping W∗ is then used to align monolingual word embeddings in a single space and the semantic edge weights are obtained based on the similarity of aligned multilingual word embeddings. Figure 2 illustrates the process of creating and weighting semantic relationships. Nevertheless, initial experiments indicate that directly combining sequential and semantic edge weights lead to worse results, due to the fact that the edge weights come from different distributions. Therefore, instead of directly using the cosine similarity, we propose to use the exponential of a scaled cosine similarity, which significantly increases the values of semantic weights and enhances the importance of transfer learning across languages. Formally, the weight of semantic edge of a word pair wi, wj is calculated as follows:

(7) weightsemantic=exp⁡(α.cosine_sim(wi,wj))

(8) cosine_sim(wi,wj)=xi⋅xj∥xi∥∥xj∥

where α is a parameter for scaling the similarity between a pair of words, and xi, xj denote respectively the aligned word embeddings of the words wi and wj. To reduce the complexity, we use bilingual dictionaries (Conneau et al., 2017) to generate a set of linkages across languages rather than computing the similarity between the vocabulary of different languages to select the most similar words.

Figure 2 The process of generating the weights of semantic relations.

A linear mapping W∗ is learned to align monolingual word embedding spaces. Then, similarity between words are computed in the aligned word vector space to generate the weights of semantic edges.

Slightly deep graph convolutional network

The vanilla graph convolutional network (GCN) (Kipf & Welling, 2016) is a graph neural network which follows a message-passing mechanism. GCN operates directly on a graph to learn node representations by aggregating the feature information from their topological neighbors. Formally, consider an attributed graph G=(V,E,A) where V is the set of n=|V| nodes, E is the set of edges, and A∈ℝn×n is the adjacency matrix of G. For each connected pair of words wi, wj, A(i,j) is the weight of the edge; otherwise, A(i,j)=0. In addition, each node is assumed to be connected to itself, i.e., ∀wi∈V;(wi,wi)∈E and A(i,i)=1. The normalized symmetric adjacency matrix of A is A~=D−12AD−12, where D denotes the degree matrix of A, and Dii=ΣjAij. Let X∈ℝn×d be a feature matrix containing the embedding vectors of all nodes of G, where d denotes the dimension of the embedding vectors and each row xi of the matrix represents the embedding vector associated to the node i. The vanilla GCN model f(X,A) learns the nodes embeddings based on two-layers as follows:

(9) f(X,A)=softmax(A~ReLU(A~XW(0))W(1))

here W(0)∈ℝd×h and W(1)∈ℝh×c are the weight matrices of the first and second layer, respectively, with h denotes the dimension of the hidden representation, and c is the number of classes. The activation functions are defined as follows:

(10) ReLU(x)=max(x,0)

(11) softmax(xi)=exp⁡(xi)Σjexp⁡(xj)

Inspired by the success of the vanilla GCN, we propose a slightly deep graph convolutional network which learns deeper and better node representations. We reformulate the vanilla GCN by increasing the depth of the architecture and adding a single feed forward neural network layer to aggregate information from beyond local neighborhoods and to learn long-range dependencies. Regarding the activation function, we adopt tanh instead of ReLU, because it makes the learning easier, as it is zero centered and its values lie between −1 to 1. However, the limitation faced by ReLU is the dying ReLU problem which decreases the learning ability of the model. Formally, our forward slightly deep graph convolutional network model is described as follows:

(12) H(1)=tanh(A~H(0)W(0))

(13) H(2)=tanh(A~H(1)W(1))

(14) H(3)=softmax(H(2)W(2)+b)

where W(0)∈ℝd×h and W(1)∈ℝh×h denote the weights of the first and second layer, respectively, W(2)∈ℝh×c is the weight matrix of the dense layer and H(0)=X. H(i) is the learned representations of nodes in the layer i. The tanh activation function, is defined as:

(15) tanh(x)=exp⁡(x)−exp⁡(−x)exp⁡(x)+exp⁡(−x)

To measure the classification performance of the slightly deep graph convolutional network, we adopt the cross-entropy loss across all labeled documents:

(16) L=−∑d∈YD∑c=1CYdclnHdc(3)

where YD is the set of labeled document indices, and C is the number of classes.

Experiments

In this section, our focus turns to the experiments performed on various datasets, aimed at assessing the performance of the MSA-GCN approach. The objective of these experiments is twofold: firstly, to investigate the effectiveness of the slightly deep graph convolutional network in acquiring predictive representations, and secondly, to explore the approach’s resilience in face of language variations.

Datasets

Throughout the experimental analysis, we adopt four datasets to evaluate the effectiveness of the proposed approach. These datasets include Multilingual Amazon Reviews Corpus (MARC) (Keung et al., 2020), Internet Movie Database (IMDB) (Maas et al., 2011), Allociné (Théophile Blard, 2020), and Muchocine (Mata et al., 2008). MARC is an extensive and diverse collection of Amazon reviews for multilingual text classification. The corpus comprises reviews from six languages: English, Japanese, German, French, Chinese, and Spanish. These reviews were gathered precisely during the period from 2015 to 2019. For each language, the training, development, and test sets consist of 200,000, 5,000, and 5,000 reviews, respectively.

IMDB is a movie reviews dataset. It consists of reviews derived from the IMDB, a popular online database of movies, television shows, and related content. It is a binary sentiment classification dataset written in English. It is made up of 25,000 highly polar movie reviews for training and another 25,000 for testing.

Allociné is a binary French-language sentiment analysis dataset which consists of movie reviews authored by Allociné website users, spanning from 2006 to 2020. These reviews cover various films and are categorized into positive and negative sentiments. In total, there are 100,000 positive and 100,000 negative reviews, which are further divided into three subsets: 160,000 for training, 20,000 for validation, and another 20,000 for testing.

Muchocine is Spanish-language sentiment analysis dataset. The dataset comprises movie reviews written by Muchocine website users about various movies. Each movie in the dataset is associated with a longform movie review, the summary review, and the rating on a 1–5 scale. The dataset contains a total of 3,872 movie reviews.

Due to the syntactic structural disparities across the six languages, we consider reviews from four languages namely, English, German, French, and Spanish. Furthermore, we reduce the problem to binary classification. So, we assigned one and two stars to the negative class, four and five stars to the positive class, and three stars to the neutral class. Then, we construct 10 datasets for Amazon reviews based on different language combinations (EN-FR, EN-ES, EN-DE, FR-ES, FR-DE, ES-DE, EN-FR-ES, EN-FR-DE, EN-ES-DE, FR-ES-DE), and one for movie reviews by combining the three movie reviews datasets, IMDB, Allociné, and Muchocine.

Tables 1 and 2 show respectively the number of documents in training, validation, and test sets and the summary statistics of movie reviews dataset and Amazon reviews datasets. In addition, Tables 3 and 4 illustrate respectively a sample of Multilingual Amazon Reviews Corpus (MARC) and the constructed movie reviews dataset.

Table 1 Number of documents in training, validation, and test sets for movie reviews dataset and Amazon reviews datasets.

Dataset	Training set	Validation set	Test set	
MARC (two languages combination)	20,000	8,000	8,000	
MARC (three languages combination)	30,000	12,000	12,000	
Movie reviews	10,646	2,138	2,138	

Table 2 Statistics of movie reviews dataset and Amazon reviews datasets.

Dataset	Vocab size	Max length	Min length	Average length	
EN-FR	36,598	753	2	34	
EN-ES	37,159	630	2	34	
EN-DE	47,174	658	2	37	
FR-ES	41,163	753	2	30	
FR-DE	51,867	753	2	34	
ES-DE	51,939	658	2	34	
EN-FR-ES	51,939	658	2	34	
EN-FR-DE	55,566	753	2	33	
EN-ES-DE	66,399	658	2	35	
FR-ES-DE	70,978	753	2	33	
Movie reviews	103,752	4,589	1	190	

Table 3 A sample of Multilingual Amazon Reviews Corpus.

Review-body	Review-title	Language	Label	
Followed directions, did not work as advertised.	Waste of money	en	neg	
Produit conforme à la photo	Bon produit	fr	pos	
Llega tarde y co la talla equivocada	Devuelto	es	neg	
Qualitativ gut verarbeitet. Einfache Anwendung.	Gutes Produkt	de	pos	

Table 4 A sample of the constructed movie reviews dataset.

Review	Lang	Label	
Au départ ça part sur un film anti militariste assez efficace puis les personnages deviennent plus complexes et l’ensemble devient plus difficile à saisir. Un casting intéressant	fr	neg	
If there is a movie to be called perfect then this is it. So bad it wasn’t intended to be that way. But superb anyway… Go find it somewhere. Whatever you do… Do not miss it!!!	en	pos	
un filme tristísimo donde el zombie es utilizado como una enfermedad terminal que acaba con la vida de su protagonista cada minuto se siente el calvario gracias a un sobresaliente giles aspen aunque no pierda la oportunidad del gore todo es un segundo plano importando el personaje principal su drama personal y sus pensamientos un filme innovador en el subgénero que incomprensiblemente no se le considera como lo mejor que el horror dio en los y tal vez uno de los mejores filmes sobre el fenómeno zombie andrew parkinson y su actor protagonista merecieron más suerte a reivindicar	es	pos	

Baseline models

We compare the proposed approach against several baselines and state-of-the-art methods. The baselines include: Bi-LSTM Emb-Non-Static (Bi-LSTM-NS): a bi-directional LSTM (Liu, Qiu & Huang, 2016) with trainable word embeddings. Bi-LSTM represents the whole text with the last hidden state. The embeddings are initialized randomly and fine tuned during the training process.

CNN Emb-Non-Static (CNN-NS): a convolutional neural network (Kim, 2014) with trainable word embeddings. The convolution is applied over the randomly initialized word embeddings to learn the word and text representations.

Bi-LSTM Emb-Static (Bi-LSTM-S): a bi-directional LSTM (Liu, Qiu & Huang, 2016) with randomly initialized word embeddings. The word embeddings are not optimized during the training process.

CNN Emb-Static (CNN-S): a convolutional neural network (Kim, 2014) with randomly initialized word embeddings. The word embeddings are not updated throughout the training process.

Char-CNN: a small version of character-level convolutional network for text classification (Zhang, Zhao & LeCun, 2015). The designed architecture extracts efficient representation based on the character features.

fastText: a simple baseline method for text classification (Joulin et al., 2017). The method averages the word features to construct efficient text representation that will be used for learning a linear classifier.

Text level GCN (TL-GCN): text level graph neural network for text classification (Huang et al., 2019). The method uses a message passing mechanism to learn representation for each input text modeled with a graph.

Graph-GCN: the vanilla GCN (Kipf & Welling, 2016), applied to the constructed heterogeneous text graph.

Experiment settings

In our experiments, we employed some common language-agnostic preprocessing techniques. These techniques involved removing URLs, eliminating special characters, converting text to lowercase, and removing HTML tags. The text was then tokenized using single space.

To optimize the proposed MSA-GCN approach, we conducted a series of experiments to determine the best hyperparameters. These experiments involve defining a grid of hyperparameter values and searching through all possible combinations of these values to find the best set of hyperparameters. Based on the experimental results we set the hyperparameters as follows: Low frequency terms: we remove terms that appeared less than five times.

Size of the sliding window: 25. We investigate different sizes of sliding window (see Fig. 3) and we judge that the best sliding window size is 25, as the small window size may not provide enough global word co-occurrence information, whereas large window size could introduce extra edges between words that are not closely connected. As observed in Yao, Mao & Luo (2019), our initial experiments demonstrate that minor adjustments to the window size have minimal impact on the results.

Word embeddings initialization: identity matrix. That is to say that each node is represented with a one-hot vector.

α: 7. We explored various values of α (refer to Eq. (7)) and determined that the most suitable value is 7 (refer to Fig. 4).

Node embedding dimension: 200 for both layers of the slightly deep graph convolutional network.

maximum of training epochs: 200.

Optimizer: Adam optimizer (Kingma & Ba, 2015).

Learning rate: 0.002.

Stop training criteria: if the validation loss does not decrease for five consecutive epochs.

Figure 3 Test accuracy of the MSA-GCN approach on the EN-FR dataset based on different sliding window sizes.

Figure 4 Test accuracy of the MSA-GCN approach on the EN-FR dataset based on different values of the parameter alpha.

For the baseline models, we use the default hyper-parameter for fastText, TL-GCN, and Char-CNN as described in their original articles. These hyper-parameters were chosen because they consistently delivered optimal performance across diverse datasets in their original research (Zhang, Zhao & LeCun, 2015; Joulin et al., 2017; Huang et al., 2019). However, we performed hyper-parameter tuning to choose the best hyper-parameter for Bi-LSTM-NS and CNN-NS. So, the results were 100 for the embedding dimension and number of units for Bi-LSTM-NS, 0.5 for dropout, and L2 for the layer weight regularizer with a weight of 0.01. Also, 100 for the embedding dimension and number of filters for CNN-NS, while the kernel size is 3. We use the same settings for Bi-LSTM-S, CNN-S as Bi-LSTM-NS and CNN-NS respectively. All four models are trained using the Adam optimizer with a learning rate of 0.001. For all baseline models we utilize randomly initialized word embedding rather than pre-trained word embedding because we explore these models in multilingual settings.

The proposed approach is implemented based on Python and PyTorch 1.11.0+cpu. The experiments are conducted on a machine with an Intel(R) Xeon(R) Platinum 8270 CPU @ 2.70 GHz and 200 GB RAM.

Results analysis

A comprehensive experiment is conducted on several datasets. We evaluate each model 30 times based on the accuracy, and we report the mean ± standard deviation. Tables 5 and 6 show the results of the MSA-GCN approach against diverse baseline models. The obtained results clearly demonstrate that the suggested approach significantly outperforms all the baselines (p-value < 0.05 based on student t-test) on eight datasets over 11, which prove the effectiveness of the proposed approach.

Table 5 Test accuracy on the multilingual document sentiment analysis task.

Model	EN-FR	EN-ES	EN-DE	FR-ES	FR-DE	ES-DE	EN-FR-ES	EN-FR-DE	EN-ES-DE	FR-ES-DE	
Bi-LSTM-NS	84.99±0.72	84.60±1.09	84.97±1.07	86.12±0.97	86.51±0.44	85.76±1.15	84.97±1.32	85.21±1.29	85.48±0.78	86.54±0.40	
CNN-NS	84.72±0.28	84.50±0.18	85.05±0.22	85.87±0.37	86.33±0.25	86.06±0.28	84.18±0.12	84.75±0.18	84.61±0.12	85.32±0.15	
Bi-LSTM-S	64.42±2.41	65.01±1.84	61.83±2.02	67.11±2.40	64.55±2.12	65.24±2.86	64.57±1.23	63.66±1.54	63.90±1.30	65.30±1.96	
CNN-S	73.52±0.69	73.26±0.98	72.89±0.86	75.49±0.51	75.00±0.72	75.07±0.77	73.84±0.89	72.99±0.79	73.12±0.97	75.30±0.66	
Char-CNN	59.67±4.88	55.03±2.26	55.04±1.43	58.55±5.33	56.30±2.48	56.16±2.16	60.06±3.27	57.15±3.09	52.30±2.76	54.12±4.20	
fastText	65.76±4.06	63.82±3.80	64.09±3.81	65.68±3.68	65.96±5.88	66.33±4.18	66.97±4.32	69.24±5.73	66.24±5.19	65.82±4.85	
TL-GCN	83.73±0.14	83.88±0.28	84.52±0.10	84.56±0.22	85.38±0.17	85.58 ± 0.12	83.30±0.12	83.63±0.10	83.81±0.11	84.27±0.11	
Graph-GCN	84.50±0.16	82.69±6.17	84.66±0.16	84.86±0.91	85.62±0.14	85.26±0.13	84.66±0.14	85.22±0.13	84.74±0.09	85.51±0.13	
MSA-GCN	85.79 ± 0.12	85.00 ± 0.15	85.68 ± 0.12	86.29 ± 0.13	87.03 ± 0.13	86.48 ± 0.15	85.80 ± 0.13	86.25 ± 0.11	85.70±0.12	86.64±0.13	
Note:

We evaluate each model 30 times and report the mean ± standard deviation. The MSA-GCN significantly outperforms the baseline models on seven datasets over 10 datasets based on student t-test (p-value < 0.05). Bold values indicate that the approach significantly outperforms the baseline models in this dataset based on student t-test (p-value < 0.05).

Table 6 Test accuracy on the multilingual movie review dataset.

Model	Accuracy	
Bi-LSTM-NS	78.29±1.57	
CNN-NS	79.78±0.92	
Bi-LSTM-S	53.24±0.05	
CNN-S	63.99±1.40	
Char-CNN	54.02±0.57	
fastText	67.68±3.66	
TL-GCN	82.86±1.08	
Graph-GCN	87.55±0.26	
MSA-GCN	88.14±0.26	
Note:

We evaluate each model 30 times and report the mean ± standard deviation. The achieved results show the superiority of the MSA-GCN over all the baseline methods. Bold values indicate that the approach significantly outperforms the baseline models in this dataset based on student t-test (p-value < 0.05).

For more detailed performance analysis, we notice that the proposed approach consistently outperforms all baseline models in almost all datasets. On the other end of the spectrum, Char-CNN achieved the poorest results across almost all datasets. This discrepancy can be attributed to the limitation of encoding multilingual text using just character features, which fails to capture meaningful information due to syntactic variations between languages. The Bi-LSTM-S and CNN-S models achieved moderate results. This can be attributed to the strong interdependence between the learned document representation and word embeddings. Conversely, random word embeddings lack the ability to capture syntactic or semantic information, rendering the task of representation learning exceedingly challenging. Furthermore, our analysis uncovers that FastText’s performance is consistently modest across all datasets. This observation underscores the limitations of forming insightful document representations for MSA through a simple averaging of word embeddings. Remarkably, CNN-S outperforms Bi-LSTM-S by a considerable margin due to its aptitude for extracting local and position-invariant features, which are critical for sentiment analysis. Also, we observe that Bi-LSTM-NS and CNN-NS obviously beat Bi-LSTM-S and CNN-S, respectively, since they can learn valuable word representations in a supervised manner.

The graph-based methods TL-GCN and Graph-GCN perform quite well, and show competitive performances, indicating the efficacy of the graph-based approaches. This suggests the efficiency of modeling the entire multilingual corpus with a single graph and its ability to capture the relations across words and documents. Also, modeling each document with a graph allows to capture the relations between words by learning more expressive edges. The proposed approach consistently outperforms TL-GCN in all datasets, indicating the effectiveness of the way proposed for constructing the graph from the entire multilingual corpus. We also observed that the proposed approach outperforms Graph-GCN, which proves the performance of the slightly deep graph convolutional network to learn efficient representation for words and documents better than the vanilla GCN.

Ablation study

We conduct several ablation studies to further investigate the contribution of each component of MSA-GCN approach, and the results are shown in Figs. 5–8.

Figure 5 Test accuracy of the original MSA-GCN approach and MSA-GCN without semantic edges on six datasets.

Figure 6 Test accuracy of the original MSA-GCN approach and MSA-GCN with normalized word-word edges on six datasets.

Figure 7 Test accuracy of the original MSA-GCN and three of its variants on six datasets, each employing distinct mutual information metrics.

Figure 8 The relationship between test accuracy of MSA-GCN and the number GCN layers on six datasets.

Semantic relationship

In this study, we compare the original MSA-GCN approach constructed as detailed in “The Proposed Approach”, with MSA-GCN without building the semantic edges. Figure 5 reveals that eliminating the semantic edges makes MSA-GCN perform slightly worse on four of the six datasets, which highlights the importance of semantic edges. This is supported by the fact that the semantic edges bridge the gap between languages and help to create a unified, interconnected graph. In addition, semantic edges across languages encourage the transfer learning which enhance the performance of learning efficient predictive representations.

Normalized weights

In order to gain a deeper insight into how well the proposed approach performs in calculating the semantic weights (refer to Eq. (7)), we conduct a comparative analysis between the original method MSA-GCN and an adapted version of MSA-GCN that utilizes normalized word-word edge weights. Regarding the adapted version of MSA-GCN, we use the normalized pointwise mutual information (NPMI) to calculate the weights of the sequential edges and we utilize the cosine similarity to compute the weights of the semantic edges. From Fig. 6, we can observe that MSA-GCN with weights calculated with NPMI and cosine similarity achieve worse results compared with the original MSA-GCN. This indicates the strength of our proposed method to compute the weights of semantic edges against the normalized weights.

Mutual information

To explore the sequential connections within the MSA-GCN approach, we conducted a comparative analysis using six different datasets. Our investigation involved comparing the original MSA-GCN, which employs positive pointwise mutual information (PPMI) for determining sequential edge weights, against two of its modified versions that utilize distinct mutual information metrics: mutual dependency (MD) and log-frequency biased mutual dependency (LFMD) (Thanopoulos, Fakotakis & Kokkinakis, 2002). The results, depicted in Fig. 7, clearly demonstrate the superiority of the original MSA-GCN over its two variant counterparts. These findings underscore the effectiveness of PPMI in modeling sequential relationships within the entire corpus when contrasted with MD and LFMD.

Number of GCN layers

Figure 8 illustrates the performance of the proposed MSA-GCN approach using different numbers of GCN layers on six datasets. Across all these datasets, the MSA-GCN approach exhibits optimal performance when employing two GCN layers. However, when restricting the approach to just one GCN layer, there is a noticeable decline in performance. This fact can be attributed to the inherent limitation of a single GCN layer in effectively capturing predictive node representations from the features of the first order neighborhoods. In addition, it is important to note that the performance of the proposed approach experiences a significant decline as the number of GCN layers increases. Frequently, the excessive depth of GCN layers can result in node representations that converge to become nearly identical across the entire graph. This phenomenon is referred to as “oversmoothing” or “oversquashing” (Wu et al., 2021).

Discussion

The results of the experiments indicate that the MSA-GCN approach is proficient in acquiring predictive representations for both words and documents. Additionally, it demonstrates a high level of performance in multilingual sentiment analysis. Two primary factors account for the superiority of MSA-GCN. Firstly, the method we introduce involves modeling the entire corpus through a unified heterogeneous graph, incorporating semantic, sequential, and statistical information. This method allows the construction of a graph that captures the global information about word relations in the whole corpus and provides a structure that enables the transfer learning across languages. In addition, it facilitates the propagation of sentimental information among documents through the connections established by words, which helps predict sentiment of documents with low-resource languages and documents with scarce words. Furthermore, it bridges the linguistic gap by leveraging semantic connections between languages. Secondly, the slightly deep graph convolutional network enables the learning of deep predictive embeddings for both words and documents by propagating feature information from neighboring nodes. This propagation allows each node to gather information not only from its direct neighbors but also from nodes that are further away in the graph. This enables the network to capture both local and global context which considerably increases the multilingual sentiment analysis performance.

Although the proposed approach was tested on the problem of multilingual sentiment analysis, it will serve as a powerful approach for tackle other natural language processing such as named entity recognition in a multilingual settings.

Even though the MSA-GCN produced efficient results and outperformed the state-of-the-art models, it has limitations. On one hand, MSA-GCN builds a single graph for the entire corpus, meaning that the size of the graph scales with the corpus size, resulting in excessive memory usage in big corpora. On the other hand, MSA-GCN is intrinsically transductive. The transductive property implies that the model’s applicability is restricted to the specific corpus on which it was initially trained. Therefore, MSA-GCN cannot support the online testing, as the structural composition of the graph and its associated parameters are intricately related to the characteristics of the original training corpus. This restriction restrains the model’s ability to generalize effectively to new and unseen data, limiting its utility in dynamic and evolving environments where adaptability is crucial. Another significant limitation of the MSA-GCN is that it does not consider the aspect of word ordering, either on the graph construction or the development of the slightly deep graph convolutional network, which brings more insights to the analysis.

Conclusion and future works

In this work, we present a graph-based approach for multilingual sentiment analysis named MSA-GCN. The MSA-GCN approach models the entire multilingual sentiment analysis corpus with a single comprehensive heterogeneous text graph based on semantic, sequential and statistical information. Through this graph, a slightly deep graph convolution network is employed to learn predictive node representations.

We conducted extensive experiments to evaluate the performance of the proposed approach across diverse datasets involving various language combinations, as well as to determine the best hyperparameters. Each model underwent 30 evaluations, and the results were reported as mean ± standard deviation. The findings clearly demonstrate the superior performance of the proposed approach, significantly outperforming all baselines (p-value < 0.05 based on student t-test) on eight out of 11 datasets, validating its effectiveness. Additionally, the results indicate that Bi-LSTM-NS achieves competitive outcomes compared to MSA-GCN across all MARC datasets. This is attributed to the fact that Bi-LSTM explicitly models consecutive word sequences in both directions, a feature not considered by MSA-GCN, where word ordering holds crucial importance in sentiment analysis.

The proposed approach provides an important innovation in modeling the whole multilingual sentiment analysis corpus with a single heterogeneous text graph which allows to capture long-distance semantics based on the global word co-occurrence and bridge the gap between languages. In addition, it provides an effective methodology based on graph convolutional network to learn predictive representations. This approach can be very useful for the challenges that persist within social media content, as the users then express themselves with a variety of languages. In addition, it will be very helpful to address the problem of code-mixed texts.

Future work concerns the generalization of the MSA-GCN to inductive settings as it is intrinsically transductive. Also, we will work on the optimization of the memory consumption in large corpora as the size of the graph scales with the size of the dataset. Furthermore, we aim to address the problem of word ordering to enhance the capability of the proposed approach in learning effective representations.

Supplemental Information

Supplemental Information 1 Multilingual movie reviews dataset.

Supplemental Information 2 Sampled multilingual Amazon reviews dataset.

The authors extend their gratitude to Mr. Mohammed Bouri for his valuable assistance in creating certain visual representations.

Additional Information and Declarations

Competing Interests

Author Contributions

Data Availability

El Mahdi Mercha is employed by HENCEFORTH.

El Mahdi Mercha conceived and designed the experiments, performed the experiments, analyzed the data, performed the computation work, prepared figures and/or tables, authored or reviewed drafts of the article, and approved the final draft.

Houda Benbrahim conceived and designed the experiments, authored or reviewed drafts of the article, and approved the final draft.

Mohammed Erradi conceived and designed the experiments, authored or reviewed drafts of the article, and approved the final draft.

The following information was supplied regarding data availability:

The raw data is available in the Supplemental Files and at GitHub and Zenodo:

- https://github.com/MERCHA/Heterogeneous-text-graph-for-comprehensive-multilingual-sentiment-analysis.

- MERCHA. (2024). MERCHA/Heterogeneous-text-graph-for-comprehensive-multilingual-sentiment-analysis: Initial Release (0.1.0). Zenodo. https://doi.org/10.5281/zenodo.10450317.

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
