# Peer review of "Heterogeneous text graph for comprehensive multilingual sentiment analysis: capturing short- and long-distance semantics"

_PeerJ Computer Science, doi:10.7717/peerj-cs.1876_

## Round 0.1 · original submission · Major Revisions

Dear authors,

Thank you for submitting your article. Reviewers have now commented on your article and suggested major revisions. When submitting the revised version of your article, it will be better to address the following:


1- The abstract should present some main points for the readers, such as the main contributions, the proposed method, the main problem, the obtained results, the benchmark tests and data, the comparative methods, etc. The contribution is not properly explained in an understandable way. The abstract section should be rewritten in order to clearly state the manuscript’s main focus.

2- Each reference must be numbered consecutively in square brackets (not in parentheses) and added to a list at the end of the paper.

3- The research gaps and contributions should be clearly summarized in the introduction section. Please evaluate how your study is different from others in the related section.

4- The values for the parameters of the algorithms selected for comparison are not given.

5- The paper lacks the running environment, including software and hardware. The analysis and configurations of experiments should be presented in detail for reproducibility. It is convenient for other researchers to redo your experiments and this makes your work easy to accept. A table with parameter settings for experimental results and analysis should be included in order to clearly describe them.

6- Clarifying the study’s limitations allows the readers to better understand under which conditions the results should be interpreted. A clear description of the limitations of a study also shows that the researcher has a holistic understanding of his/her study. However, the authors fail to demonstrate this in their paper. The authors should clarify the pros and cons of the methods. What are the limitation(s) methodology(ies) adopted in this work? Please indicate practical advantages, and discuss research limitations.

7- The conclusion section is indicative, but it might be strengthened to highlight the importance and applicability of the work done with some more in-depth considerations, to summarize the findings, and to give readers a point of reference. Additional comments about the reached results should be included. It can be rewriten considering the following comments:
- Highlight your analysis and reflect only the important points for the whole paper.
- Mention the benefits.
- Mention the implications at the end of this section.

8- Reviewer 1 has requested that you cite specific references. You are welcome to add it/them if you believe they are relevant. However, you are not required to include these citations, and if you do not include them, this will not influence my decision.

**Language Note:** The review process has identified that the English language must be improved. PeerJ can provide language editing services - please contact us at copyediting@peerj.com for pricing (be sure to provide your manuscript number and title). Alternatively, you should make your own arrangements to improve the language quality and provide details in your response letter. – PeerJ Staff

Reviewer 1 ·

Basic reporting

The authors propose a multilingual sentiment analysis approach based on a graph convolution network (MSA-GCN) to capture both short- and long-distance semantics.
• The abstract would benefit from a revised structure that includes sections for the study's background, objectives, materials and methods, results, conclusions, and recommendations. It appears more like an introductory background statement that the authors should let the abstract be quantitative. The background introduction of the abstract should also be decreased.
• Your in-text citation is very wrong; you need to check the journal template for the appropriate citations. For instance, if it is a number citation as you have used you use [1], [2], [3, 4] not bracket (1), (2) as you have used so, please correct as appropriate.
• Authors should review related works in a new section, state the gaps discovered from the studies reviewed, and note how their work intends to improve on the limitations found in the literature.
• The authors should state how the study performance was accessed or evaluated.
• The study should be compared with existing systems (state-of-the-art), and authors should state how it surpassed the existing one and why it performed less or less.
• The limitation of the study should be stated, and they should present future research work.
• Source codes should be provided for replicating the study
• Overall, the English language and presentation style should be improved significantly. There were a lot of grammatical errors and typos. I suggest you have a colleague proficient in English and familiar with the subject matter review your manuscript or contact a professional editing service.
• I have suggested publications relating to your study that I want the authors to cite and reference in their papers. These are recent articles and will be of benefit to the author's manuscript and the journal as well.

a. Liu, X., Zhou, G., Kong, M., Yin, Z., Li, X., Yin, L.,... Zheng, W. (2023). Developing Multi-Labelled Corpus of Twitter Short Texts: A Semi-Automatic Method. Systems, 11(8), 390. doi: 10.3390/systems11080390
b. Liu, X., Wang, S., Lu, S., Yin, Z., Li, X., Yin, L.,... Zheng, W. (2023). Adapting Feature Selection Algorithms for the Classification of Chinese Texts. Systems, 11(9), 483. doi: 10.3390/systems11090483
c. Cheng, Y., Lan, S., Fan, X., Tjahjadi, T., Jin, S.,... Cao, L. (2023). A dual-branch weakly supervised learning based network for accurate mapping of woody vegetation from remote sensing images. International Journal of Applied Earth Observation and Geoinformation, 124, 103499. doi: https://doi.org/10.1016/j.jag.2023.103499
d. Liu, X., Shi, T., Zhou, G., Liu, M., Yin, Z., Yin, L.,... Zheng, W. (2023). Emotion classification for short texts: an improved multi-label method. Humanities and Social Sciences Communications, 10(1), 306. doi: 10.1057/s41599-023-01816-6
e. Wang, Y., Su, Y., Li, W., Xiao, J., Li, X.,... Liu, A. (2023). Dual-path Rare Content Enhancement Network for Image and Text Matching. IEEE Transactions on Circuits and Systems for Video Technology. doi: 10.1109/TCSVT.2023.3254530
f. Nie, W., Bao, Y., Zhao, Y., & Liu, A. (2023). Long Dialogue Emotion Detection Based on Commonsense Knowledge Graph Guidance. IEEE Transactions on Multimedia. doi: 10.1109/TMM.2023.3267295
g. Dong, J., Hu, J., Zhao, Y., & Peng, Y. (2023). Opinion formation analysis for Expressed and Private Opinions (EPOs) models: Reasoning private opinions from behaviors in group decision-making systems. Expert Systems with Applications, 121292. doi: https://doi.org/10.1016/j.eswa.2023.121292
h. Liu, Y., Li, G., & Lin, L. (2023). Cross-Modal Causal Relational Reasoning for Event-Level Visual Question Answering. IEEE Transactions on Pattern Analysis and Machine Intelligence, 45(10), 11624-11641. doi: 10.1109/TPAMI.2023.3284038
i. Liu, Z., Wen, C., Su, Z., Liu, S., Sun, J., Kong, W.,... Yang, Z. (2023). Emotion-Semantic-Aware Dual Contrastive Learning for Epistemic Emotion Identification of Learner-Generated Reviews in MOOCs. IEEE Transactions on Neural Networks and Learning Systems. doi: 10.1109/TNNLS.2023.3294636
j. Zhenfeng Liu, Jian Feng, Lorna Uden, 2023. Technology opportunity analysis using hierarchical semantic networks and dual link prediction. Technovation 128, 102872. https://doi.org/10.1016/j.technovation.2023.102872

Experimental design

1. The authors should state the configuration of the experiment conducted. What are the software and hardware used for the implementation of the experiment?
2. The authors should discuss how the parameters used for the experiment were tuned and set.

Validity of the findings

No comment

Additional comments

No comment

Reviewer 2 ·

Basic reporting

1. The authors used clear and unambiguous English.
2. The intext citation should be properly done all through the work with square brackets and in chronological order.
3. The result was very clear and verifiable using the same dataset
4. Line 67 in the reference seems incomplete. Authors should take note and provide the concluding word or phrase.
5. Reference [5] numbering is missing in the references

Experimental design

1. The research is with the scope of the journal.
2. The knowledge gap was clearly stated and the research tried to fill the gap to meaningful extent
3. The method was clear

Validity of the findings

1. The findings are very good based on the used dataset.

Reviewer 3 ·

Basic reporting

no comment

Experimental design

no comment

Validity of the findings

no comment

Additional comments

This work presents a novel multilingual sentiment analysis method, named MSA-GCN, based on Graph Convolutional Networks. The method is designed to effectively capture both short-distance and long-distance semantics. Extensive experiments across diverse datasets demonstrate the method's effectiveness and its robust performance across various language combinations, showcasing its superiority in handling linguistic variations.

Strengths:
(1). Innovative Approach: Introduced a pioneering multilingual sentiment analysis method based on graph convolutional networks, effectively capturing both short-distance and long-distance semantics.
(2). Comprehensive Experiments: Conducted thorough experiments on diverse datasets, providing strong evidence of the method's effectiveness.
(3). Robust Performance: Showcased the method's robustness across various language combinations, indicating its applicability to diverse linguistic contexts.

Suggestions:
(1). Detailed Methodology and Experimental Settings: Provide more detailed information on the method's design and experimental settings to enhance the understanding of research complexity and reproducibility.
(2). Comparison with Relevant Works: Include a comparative analysis with existing works to highlight the method's innovativeness and superiority in the field.
(3). Clarity on Long-Distance Semantics: Clearly specify how the proposed method improves the capture of long-distance semantics compared to existing methods, providing specific examples or metrics.
(4). Motivation for Graph Convolutional Networks: Clearly articulate the motivation for choosing graph convolutional networks in the abstract to help readers understand the rationale behind this choice.
(5). Detailed Analysis of Graph Neural Networks: Further analyze and explain the specific reasons and motivation behind using graph neural networks, addressing the current lack of clarity on this choice.
(6). Focus on Key Aspects: Remove irrelevant sections such as 2.2 and replace them with a more detailed analysis of the current state of graph neural networks in multilingual sentiment analysis, ensuring the paper remains focused on key aspects.
(7). Recent Comparative Analysis: Compare research results with relevant works from the past three years to emphasize the method's innovation and superiority.
(8). Detailed Exploration of Innovation: Explore and describe in more detail the innovative aspects and specifics of using graph neural networks for multilingual sentiment analysis, highlighting the method's uniqueness.

Language and Structure:
(1). Grammar and Clarity: Pay attention to grammar and clarity of expression to ensure readers accurately understand key concepts.
(2). Structural Clarity: Clearly present the motivation, method, experiments, and results in the manuscript to enhance readability and comprehension.

In summary, while the proposed multilingual sentiment analysis method demonstrates excellent performance in capturing both short and long-distance semantics, strengthening the method description, comparing with related works, clarifying improvements, providing clear motivation, and exploring innovation details will significantly enhance the paper's quality and understanding.

---

## Round 0.2 · accepted · Accept

Dear authors,

Thank you for the revision and for clearly addressing all the reviewers' comments. I confirm that the paper is improved and addresses the concerns of the reviewers. Your paper is now acceptable for publication in light of this revision.

Best wishes,

Reviewer 1 ·

Basic reporting

I am pleased to inform you that, after a thorough review, all necessary corrections have been successfully made to your manuscript. The revisions and updates you have implemented significantly enhance the overall quality of the paper. Therefore, I am recommending your article for publication. The diligence and effort put into addressing the previous concerns are commendable, and the manuscript is now well-positioned to make a valuable contribution to the field.

Experimental design

No comment

Validity of the findings

No comment

Additional comments

No comment

Reviewer 2 ·

Basic reporting

Basic reporting
1. The authors used clear and unambiguous English.
2. The intext citation should be properly done all through the work with square brackets and in chronological order.
3. The result was very clear and verifiable using the same dataset
.

Experimental design

Experimental design
1. The research is with the scope of the journal.
2. The knowledge gap was clearly stated and the research tried to fill the gap to meaningful extent
3. The method was clear

Validity of the findings

Validity of the findings
1. The findings are very good based on the used dataset.

Reviewer 3 ·

Basic reporting

no comment

Experimental design

no comment

Validity of the findings

no comment

Additional comments

Authors have addressed all the comments and updated the manuscript accordingly.